# Non-linear relationships between daily temperature extremes and US agricultural yields uncovered by global gridded meteorological datasets

Dylan Hogan [1] ✉ & Wolfram Schlenker [2]

Global agricultural commodity markets are highly integrated among major producers. Prices are driven by aggregate supply rather than what happens in individual countries in isolation. Estimating the effects of weather-induced shocks on production, trade patterns and prices hence requires a globally representative weather data set. Recently, two data sets that provide daily or hourly records, GMFD and ERA5-Land, became available. Starting with the US, a data rich region, we formally test whether these global data sets are as good as more fine-scaled country-specific data in explaining yields and whether they estimate similar response functions. While GMFD and ERA5-Land have lower predictive skill for US corn and soybeans yields than the fine-scaled PRISM data, they still correctly uncover the underlying non-linear temperature relationship. All specifications using daily temperature extremes under any of the weather data sets outperform models that use a quadratic in average temperature. Correctly capturing the effect of daily extremes has a larger effect than the choice of weather data. In a second step, focusing on Sub Saharan Africa, a data sparse region, we confirm that GMFD and ERA5-Land have superior predictive power to CRU, a global weather data set previously employed for modeling climate effects in the region.

Assessing the effect of climate change on global food systems requires a global analysis of how weather and climate affect agricultural productivity. Recent studies using fine-scaled weather data for individual countries or regions have shown that temperature extremes, especially extreme heat, are a main driver of agricultural yields[1,2] and whether and how agriculture can adapt to extreme heat[3,4] has major implications for the impacts of climate change. Incorporating the full temperature distribution between the daily minimum and maximum provides much better predictions of heat-related yield losses[5,6]. Averaging over time (monthly rather than daily data) or space (larger grids) can mask this nonlinear relationship[7]. However, until recently, most global data sets have only provided monthly data that can mask daily extremes (e.g.,

CRU, University of Delaware) and global studies were forced to rely on this more aggregated monthly data[8].

Recently, two new daily data sets have become available and are used extensively. They are the Global Meteorological Forcing Dataset (GMFD)[9], which includes daily minimum and maximum temperature measurements on a 0.25° grid, and ERA5-Land[10], which provides some of the most detailed temperature data both temporally (hourly) as well as spatially (0.1°) for the entire world. Given the global coverage over all agricultural areas, these two data sets have the advantage of offering a standardized weather product, which is crucial for a unified global analysis that drives prices, comparative advantages, production, and trade[11–13]. However, these global daily weather data sets have

[1]Columbia University School of International and Public Affairs, New York, NY, USA. [2]Columbia University School of International and Public Affairs, NBER and CEPR, New York, NY, USA. ✉e-mail: dth2133@columbia.edu

not been systematically assessed in how well they explain outcomes of interest compared to more detailed fine-scaled weather data sets that are available for individual countries. Coarse measurements of daily data may have substantial measurement error that will cause attenuation bias, which is especially important for studies of non-linear effects[14]. For example, if the local daily maximum temperature is not correctly captured due to measurement error in a data set induced by averaging over space, too much (or too little, depending on the sign of the error) of the temperature exposure is counted as harmful yield-decreasing heat rather than beneficial yield-increasing moderate temperatures. In this case, the coefficients of an estimated relationship between weather and yields would be biased toward zero, given that the sign of the coefficients switches over the critical threshold between beneficial and harmful temperature days. Thus, the role of temperature extremes can only be uncovered if their exposure is correctly captured in the underlying weather data set.

To better understand whether global climate data sets can uncover the effect of extreme heat on crop yields, we conduct two analyses, one in the US and one for Sub-Saharan Africa. Starting with the United States, we estimate statistical yield-weather relationships using (1) a modified version of the fine-scaled PRISM data set[15], which only provides weather measurements in the US, and (2) the same measurements constructed from the more aggregated, but globally available, data sets, GMFD and ERA5-Land. We compare model performance across data sets for the area where they overlap, i.e., the contiguous United States. We show that all three data sets give comparable response functions and climate change predictions. Our paper focuses on agriculture, but similar response functions have been estimated for other sectors (e.g., energy[16] or mortality[17]). While agriculture accounts for only 4% of global economic activity, it comprises more than a quarter of GDP and more than half of the employment in some of the least developed countries, which are among the most exposed to extreme weather and the least equipped to invest in adaptation. Accounting for the sensitivity of yields in these locations is vital for evaluating the extent of the inequalities associated with climate change impacts.

Our statistical analysis links yields to each weather data set. Yield data are publicly available at the county level in the United States. We focus on counties east of the 100° meridian (except Florida) to avoid bias associated with subsidized irrigation systems[1]. After this restriction, 73% of all counties report corn yields, while 63% of all counties report soybean yields in at least some of the years of our 70-year sample period 1950–2019. GMFD is only available through 2010, and thus our sample for this data set shrinks to 1950–2010. Our results are robust to imposing this restriction on other data sets, with the only significant difference being a small reduction in out of sample performance for ERA5-Land models of soybean yields. We aggregate all weather data sets to the county level, weighted by the share of cropland in a cell as measured by a satellite-derived cropland mask that is averaged over the available years between 2008 and 2021 at the 30 × 30 m scale. For the fine-scaled PRISM data set (1/24° grid), we link each grid cell to the county in which the centroid is located. For the coarser ERA5-Land and GMFD data set (0.1° and 0.25°, respectively), we weight by the area of each grid cell that falls within a county. Given the importance of non-linearity in these relationships, we first derive all non-linear transformations at the daily grid-cell level before aggregating the data to counties. We construct our weather measures for March–September to ensure we cover the entire growing season.

For each data set, we estimate non-linear relationships between temperature and corn or soybean yields, controlling for precipitation, state-specific quadratic time trends (to capture technological change) as well as county fixed effect to capture all time-invariant confounding factors (e.g., soil quality). Earlier research has shown that different temperature ranges can have opposing effects, where moderate temperatures are yield enhancing while very hot temperatures greatly

reduce yields. We consider three functional form assumptions to capture possible non-linear relationships. First, we estimate piecewise linear regressions that separate the impacts of moderate days and extreme days around a critical temperature threshold, following the concept of degree days in the agronomic literature. Since this specification requires a break-point between beneficial temperatures and yield-decreasing extreme heat, we apply a data-driven cross validation approach to inform this decision separately for each crop and weather data set. Second, we estimate a flexible 8th order Chebychev polynomial, which smoothly characterizes the temperature yield relationship and does not require a threshold assumption. Third, we consider a semi-parametric specification, which estimates a separate yield effect for each three-degree temperature range.

In a second step, we conduct an analogues analysis for a data-poor region: countries in Sub-Saharan Africa. Since there is no sub-national yield data available for the region, we revert to using the remotely sensed Enhanced Vegetation Index (EVI), which has been shown to correlate well with yields. Our dependent variable is the log of total EVI over the growing season, which we aggregate to the ERA5-Land grid (0.1° grid). Unfortunately, we do not have a fine-scaled daily weather data set for Sub-Saharan Africa. We, therefore, compare ERA5-Land and GMFD to a monthly weather data set (CRU) that has been used in studies of yields in Africa[18]. CRU also differs from ERA5-Land and GMFD by statistically interpolating weather measurements across stations rather than using reanalysis to construct a complete weather record. We conduct the analysis separately for grids that have a weather station in them and grids without weather stations: In both cases, ERA5-Land and GMFD perform statistically significantly better than CRU at explaining crop yields, even in this data-sparse area.

## Results
### Relationships estimated on county-level yield data in the United States are quantitatively similar for all three functional form assumptions

We asses model performance using a cross-validation procedure that quantifies the ability of the response functions to predict crop yields outside of the estimation sample. We show that PRISM models outperform the ERA5-Land and GMFD models in predicting crop yields out of sample, particularly for soybean yields, with ERA5-Land and GMFD models performing about the same despite different temporal and spatial resolutions. While there are differences in out-of-sample prediction accuracy, we also show that spatially-uniform warming scenarios applied to the response functions project climate impacts of similar magnitude and precision.

### All specifications and weather data uncover an asymmetric relationship for the US where yields are increasing in temperature for moderate temperature ranges, but sharply decrease in temperature at the upper end

Figure 1 displays response functions for the temperature sensitivity of crop yields for all three weather data sets and 95% confidence intervals accounting for spatial correlation of errors within states. The figure has six plots, with rows examining different crops and columns different functional forms for the temperature-yield relationship. In particular, we estimate responses for two crops, corn (top row) and soybeans (bottom row) with three functional form assumptions, a piecewise linear specification following the agronomic concept of degree days (left column), an 8th order polynomial in temperature (middle column), and a step function that estimates a separate temperature effect for each 3° bin (right column). Within each plot, response functions estimated from PRISM, ERA5-Land, and GMFD data are shown in different colors. The red lines show responses based on PRISM, which provides spatially fine-scaled daily data for the contiguous United States. The blue lines show responses based on ERA5-Land, using the data for the contiguous US in this globally representative hourly data

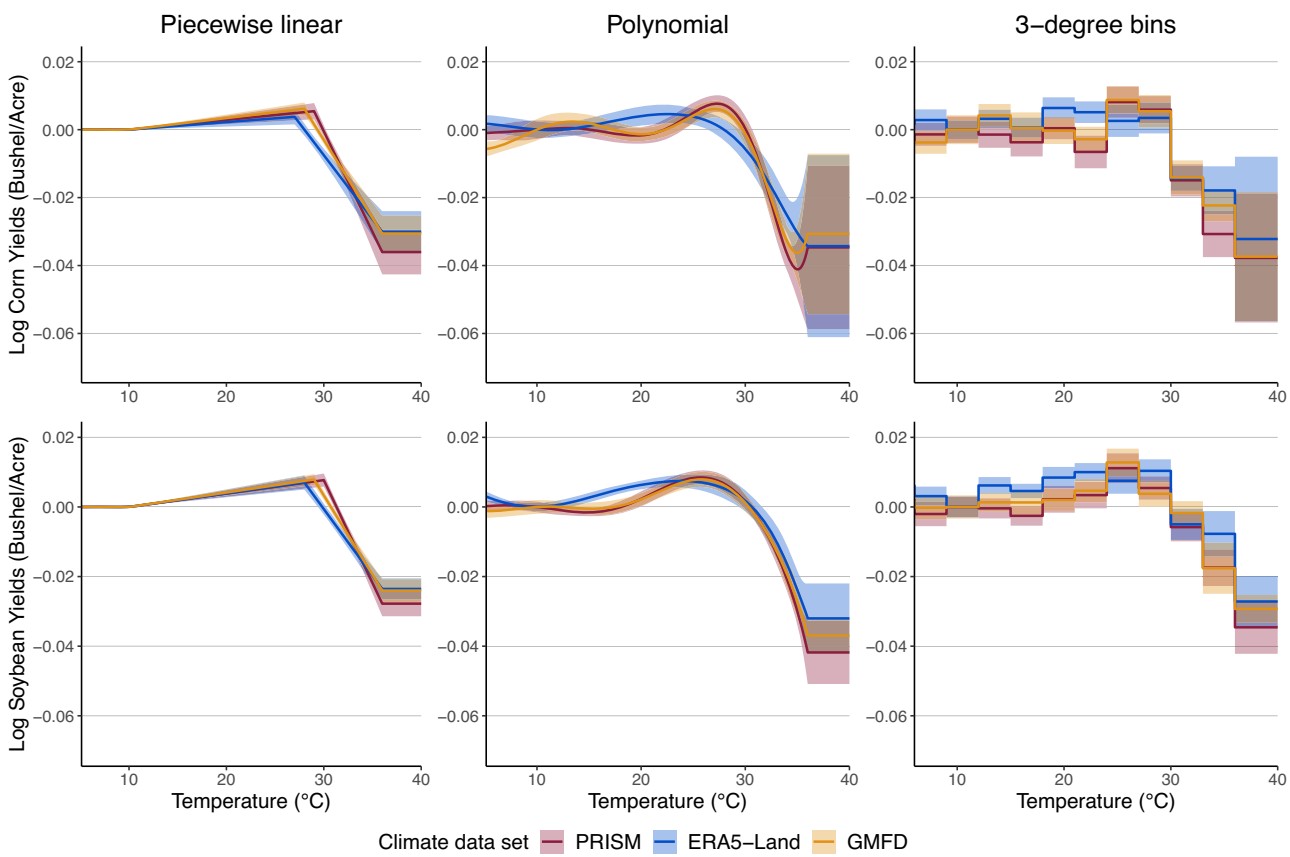

**Fig. 1 | Comparing the yield-temperature response across the three weather data sets.** Each graph estimates the relationship between US yields and temperature using both the fine-scaled PRISM data set (shown in red) as well as the more aggregate but globally available ERA5-Land (shown in blue) and GMFD (shown in yellow). Lines indicate estimated response functions from the panel regression models described in Methods and bands indicate the 95% confidence intervals. The top row provides results for corn yields, while the bottom row gives the results for soybean yields. The left column estimates a piecewise linear function, the middle column an 8th order polynomial in temperature, and the right column uses temperature bins.

set at the 0.1° resolution. The yellow lines show responses based on GMFD, again focusing on the US portion of the data for this globally representative daily data at the 0.25° resolution.

The dependent variable in all cases is the log of crop yield (i.e., bushels per acre). The response functions in Fig. 1 are normalized to zero at 10° C. Since our model includes county fixed effects, the results should be interpreted in relative terms, i.e., the difference in height ($y$-value) for two different temperatures ($x$-values) rather than in absolute levels so the normalization is inconsequential. All three climate data sets estimate similar response functions across crops and specifications, with modest yield increases associated with going from cold to moderate temperatures and sharp yield decreases once temperatures pass a threshold. Notably, the cross validation procedure we use to choose the critical temperature threshold for the piecewise linear specification yields different breakpoints for each climate data set. For corn, the threshold varies from 27 °C (ERA5-Land) to 30 °C (GMFD). For soybeans, the threshold varies from 28 °C (ERA5-Land) to 30 °C (PRISM).

Table 1 shows the effect of changes in the temperature exposure on annual crop yields. Specifically, it gives the predicted yield change (in percent) for replacing a full day at a reference 10 °C, i.e., a 24-h exposure to 10 °C (recall that we are counting partial days) with either (i) a full day at 36 °C, the hottest temperature bin estimated by our response functions, or (ii) a full day at the model-specific yield-maximizing temperature. The effect of extreme heat on crop yields is comparable across climate data sets. Substituting a full day at 10 °C with a full day at 36 °C decreases corn yields between 3.2% ($SE = 1.2\%$) and 3.8% ($SE = 1.0\%$) for all three data sets. However, models disagree on the benefits of being at the yield-maximizing temperature. Substituting a full day at a 10 °C for a full day at ($SE = -0.29\%$) and 0.87% ($SE = 0.23\%$) based on PRISM and GMFD relationships, but only 0.64% ($SE = 0.16\%$) according to ERA5-Land. For soybeans, an additional full day at 36 °C reduces yields by 3.5% ($SE = 0.38\%$) for PRISM, which is slightly larger in magnitude compared to the global data sets. A full day

**Table 1 | Crop yield sensitivity to various temperatures**

| Yield relative to a full day at 10 °C day | PRISM | ERA5-Land | GMFD |
|---|---|---|---|
| Corn yields | | | |
| 36 °C | −3.77 | −3.22 | −3.74 |
| | (0.98) | (1.20) | (1.05) |
| Yield-maximizing temperature | 0.81 | 0.64 | 0.87 |
| | (0.29) | (0.16) | (0.23) |
| Soybean yields | | | |
| 36 °C | −3.46 | −2.72 | −2.93 |
| | (0.38) | (0.32) | (0.23) |
| Yield-maximizing temperature | 1.12 | 1.04 | 1.28 |
| | (0.23) | (0.20) | (0.19) |

Table provides the predicted change in yields, in percent, associated with replacing a day (24-h exposure) at a reference 10 °C with (1) a day at 36 °C (i.e., the hottest temperature bin of our response functions) and (2) a day at the yield-maximizing temperature, which varies based on the climate data set. Standard errors are provided in parentheses.

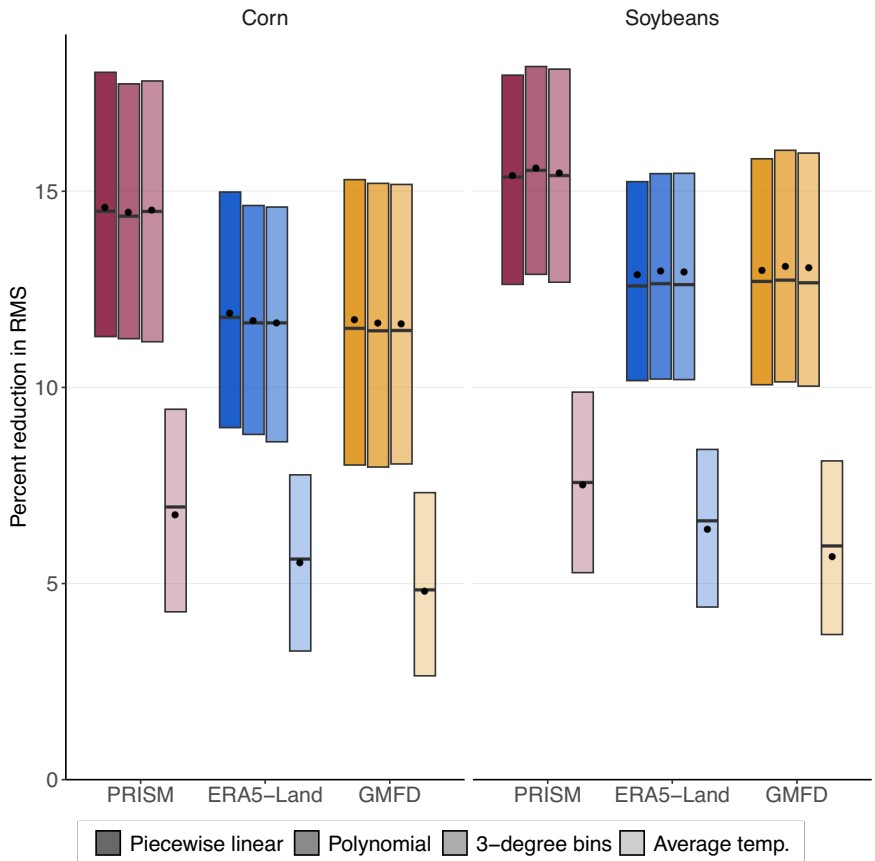

**Fig. 2 | Out-of-sample model predictions across specifications and weather data sets.** Figure compares out-of-sample prediction for piecewise linear regression models estimated using weather observations from PRISM (red), ERA5-Land (blue), and GMFD (yellow) data sets. Box shading indicates the functional form of temperature, including piecewise linear, 8th order polynomial, 3-degree bins, and growing season average temperature. The vertical axis shows the percent reduction in root-mean-squared error (RMS) relative to a baseline model that excludes all weather variables. A value of 0 zero implies the weather variable cannot explain any of the year-to-year variation in yields around the trend, while a value of 100 implies that the weather can explain the entirety. For each data set, response functions are estimated 1000 times, each time randomly sampling 85% of the years from the full panel. RMS is calculated based on each model's prediction of the remaining 15% of years. Boxes, horizontal lines, and points represent the interquartile range, median, and mean of RMS reductions from the 1000 draws, respectively. Results are provided for corn (left panel) and soybeans (right panel) yields, and climate data sets are indicated by color, while the shading of a color refers to the functional form of the temperature variables. All specification include a quadratic in season-total precipitation.

at a yield-maximizing temperature increases soybean yields between 1.0% ($SE = 0.20$%) and 1.3% ($SE = 0.19$%).

All regression models include county fixed effects and two additional sets of controls: a quadratic function of total growing season precipitation and state-specific quadratic time trends. The precipitation control features a similar concave shape with statistically significant linear and quadratic terms for all crops and climate data sets, indicating that moderate precipitation levels are optimal for yields (see Supplementary Fig. 2). State-specific quadratic time trends control for technological progress common to counties within a state. The temperature sensitivity of crop yields is robust to changes in the time trend, e.g., by including more flexible controls like year fixed effects to account for common year-specific shocks (see Supplementary Datasets 1 and 2).

### The fine-scaled PRISM data have the best out-of-sample model performance for US yields

We compare regression models from each data set using a cross validation procedure that calculates the root-mean squared prediction error (RMS) of out-of-sample predictions. In particular, we measure the reduction in RMS relative to a baseline model that includes county fixed effects and state-level quadratic time trends but excludes the weather variables. We thereby measure how much of the error can be explained by the inclusion of various weather variables. The cross

validation procedure consists of 1000 repetitions in which we randomly use 85% of the years in our panel data set in the estimation of the regression coefficients and then predict crop yields for the remaining 15%. We sample years rather than observations to avoid significant spatial correlation in yields across counties within a year.

Figure 2 shows comparisons of RMS reductions across climate data sets for corn and soybeans, with PRISM indicated by the red bar, ERA5-land indicated by the blue bar, and GMFD indicated by the yellow bar. On average across our three main specifications (piecewise linear, 8th order polynomial, and 3-degree bins), PRISM improves upon the model without weather by 14.9%, which is a significantly larger reduction in RMS (almost 3 percentage point) than ERA5-Land (12.1%) and GMFD (12.0%), i.e., the reduction is a fourth higher. For soybean yields, PRISM reduces RMS by 15.7% on average, which is again almost 3 percentage points larger than ERA5-Land (13.1%) and GMFD (13.2%), or an improvement by roughly one fifth. Note that for both crops, Welch tests[19] find a statistically significant differences in out-of-sample performance between PRISM and either of the two global data sets ($P < 0.01$ for all daily specifications and crops), however, a statistically insignificant difference between the two global data sets ERA5-land and GMFD ($P > 0.1$ for all daily specifications and crops). Consistent with prior work[1,5], the three main specifications, which leverage daily temperature extremes, outperform models that include quadratic functions of the average temperatures over the growing season, shown

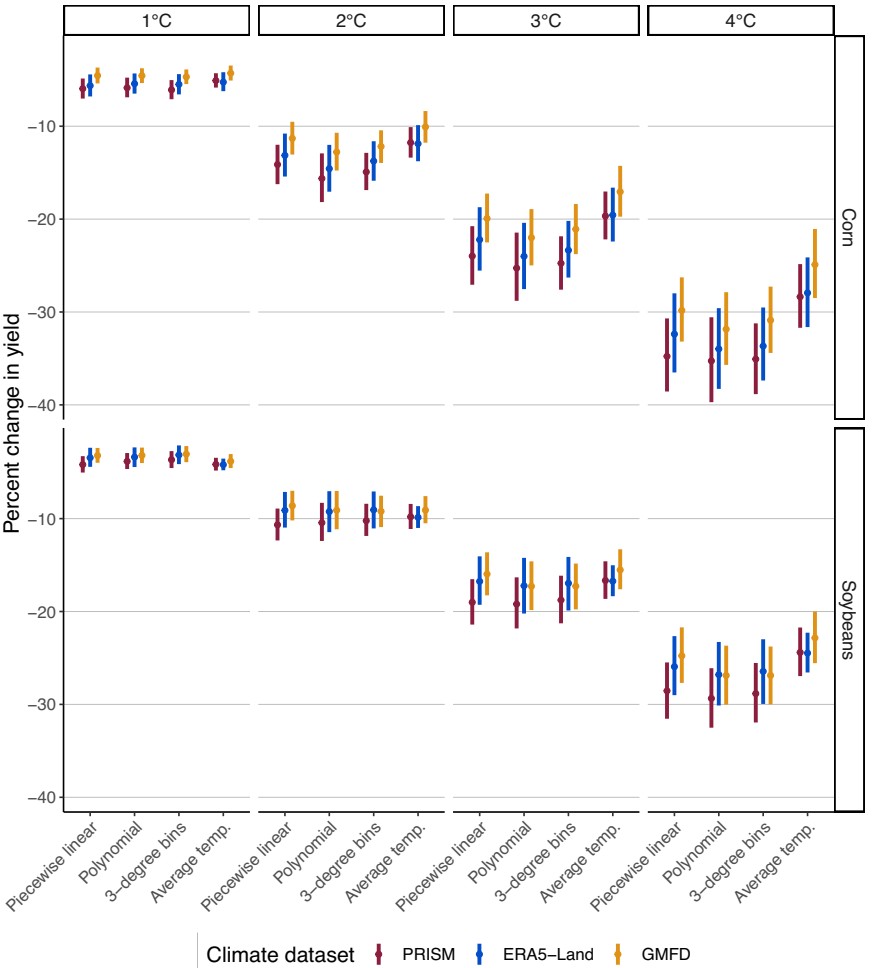

**Fig. 3 | Simulating yield losses for a range of uniform warming scenarios.** The four panels in each row provide projected climate change impacts on crop yields under uniform warming scenarios between 1 °C and 4 °C. Each panel provides the point estimates (circle) and 95% confidence bands (vertical bars) for twelve estimates when each of the four specifications are paired with the three weather data sets. Point estimates represent climate change impacts projected from the estimated parameters of the response functions and the confidence intervals are projections from 1000 draws of the statistical uncertainty of the response functions. The four specifications are listed on the horizontal axis: piecewise linear, eighth-order Chebychev polynomial, 3-degree bins, and a quadratic in average temperature. Colors indicate the climate data set used to estimate the response functions (PRISM in red, ERA5-Land in blue and GMFD in yellow). The top panel shows projected impacts on log corn yields, while the bottom panel displays the results on log soybean yields. The vertical axis is the predicted decline in overall US yields in percent.

as last column in each panel, where the RMS reduction drops by more than half. This is even true across models: the RMS reduction is larger using any of the three weather data sets or non-linear specifications than when using average temperatures. Accounting for daily extremes consistently improves model fit.

**Predicted declines in US yields are similar in terms of magnitude and precision across all climate data sets for the models using daily temperature extremes, but diverge for the model using a quadratic in average temperatures**

We compare climate change projections across historical climate data sets with uniform shifts of the temperature distribution between 1 °C and 4 °C. Many studies have used global climate models (GCMs) to project agricultural losses of spatially heterogeneous climate change scenarios, such as those from CMIP5 or CMIP6. However, errors in GCM output can be large relative to historical data sets[20] and must be bias corrected to a particular historical data set before climate impacts are calculated. There is also disagreement between GCM models on the magnitude of warming. We therefore opt for simple uniform warming scenarios that allow us to compare across data sets and to highlight the range of warming at which the models start to diverge.

Figure 3 summarizes projected yield losses, in percent, from uniform warming scenarios between 1 °C and 4 °C, including point estimates and 95% confidence intervals derived from 1000 draws from the statistical uncertainty of the response functions estimated for each data set, temperature specification, and crop. The vertical axis is the percent reduction in aggregate US crop yields relative to average yields between 1960 and 1989 after a uniform shift in the temperature distribution, and the horizontal axis indicate the four model specification we used: the first three use daily data, while the fourth (quadratic in average temperature) relies on the seasonal average. The top and bottom frames show projected impacts for corn and soybean yields, respectively.

While some counties experience yield benefits from moderate warming scenarios, negative aggregate impacts across all specifications, crops, and warming scenarios are driven by increases in the number of extremely hot days (see Supplementary Fig. 3 for maps of county level impacts). Disagreement between climate data sets regarding the threshold between beneficial and harmful temperature days drives modest differences in impacts for a given main specification and crop. Across the first three specifications and all climate data sets, corn losses from a 2 °C warming range from 11–16% and soybeans

losses range from 9–11%. A 4 °C uniform warming is projected to reduce yields between 30–35% for corn and 25–29% for soybeans. Consistent with earlier studies focusing on the functional form of temperature in climate analyses[1,5], all climate data sets agree that excluding daily variation in temperature by focusing on seasonal averages results in lower projected impact as they fail to capture the asymmetric relationship where being above the optimal temperature is much worse than being below the yield-maximizing temperature.

**An analysis using remotely sensed proxies for yields in Sub-Saharan Africa, a data-sparse region, confirms that the two global weather data sets have better predictive performance than a traditional climate data set using monthly aggregates**

One possible explanation for the high performance of global data sets in the United States is that they assimilate reliable weather station records that in turn produce better weather estimates. As such, weather may be less precisely measured in relatively data-poor regions of the world. Here, we assess the performance of the global data sets in Sub-Saharan Africa, where the network of reliable weather stations is sparse relative to the United States. There are several challenges to doing an exact replication of our US analysis in this region.

First, the weather stations available in Sub-Saharan Africa do not provide reliable daily measurements from which to calculate degree days or daily rainfall. Therefore, we do not have a "groundtruth" data set from which to compare the global data sets. We overcome this challenge by (1) identifying the location of the most reliable weather stations in Sub-Saharan Africa, and (2) comparing responses estimated with the global data sets separately for locations with weather stations in them and locations without stations. The difference in out-of-sample performance between models estimated on data with and without weather stations may provide evidence of the importance of nearby weather stations in particularly data-sparse regions. We also include a third data set in the comparison, CRU, which is a global data set that uses a statistical algorithm to impute weather station data across space rather than reanalysis. While CRU has been used to analyze the yield weather relationship in Sub-Saharan Africa[18], its drawbacks are low spatial (0.25°) and temporal (monthly) resolution.

Second, we do not have reliable sub-national yield estimates in Sub-Saharan Africa. We link monthly EVI observations with high resolution information on harvested area in Sub-Saharan Africa and a sub-regional crop calendar, aggregating over the growing season to proxy for corn yields. EVI is merged with grid-cell level measurements of degree days from ERA5-Land, GMFD, and CRU, and this 10-year panel is used to compare the out-of-sample performance of the piecewise linear relationship between log of total EVI and weather across the three data sets. Further details of this analysis are provided in Methods Section below.

The results are provided in Fig. 4. We find larger reductions in RMS for the two daily weather data sets (ERA-Land and GMFD) than the traditional monthly data (CRU). ERA5-Land has particularly high out of sample performance. ERA5-Land does not use ground-level temperature observations to estimate weather, and is instead based on satellite temperature measurements and reanalysis, which may provide a larger advantage in data-poor regions. Differences between grids with (darker shade) and without weather station (lighter shade) are not statistically significantly different for ERA5-Land and CRU, while GMFD models estimated on grids with weather station cells achieve statistically significantly better performance, though the difference remains small in magnitude. The interquartile range of the RMS estimates is skewed for all data sets regardless of the proximity of weather stations. Due to the shorter sample frame, the out-of-sample prediction subsets only include 3 years. If one of those years includes non-weather events (e.g., pest outbreaks) that had a large effect on yields, then the weather variables would reduce RMS less relative to the baseline model. The interquartile range is hence skewed. In general, while the three data

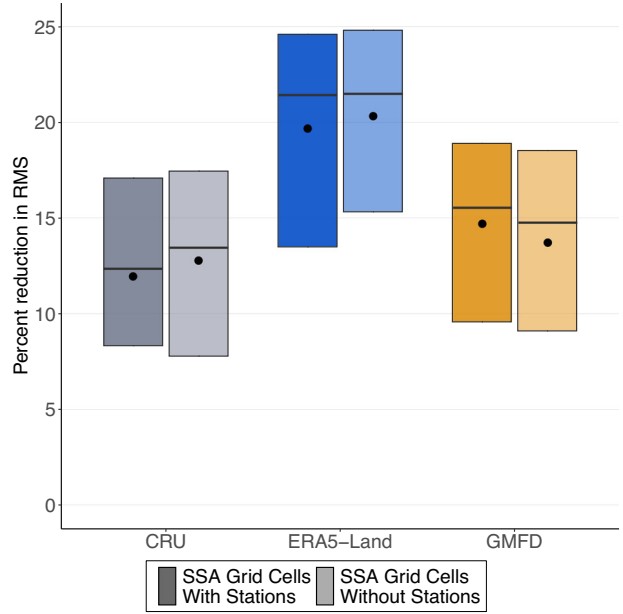

**Fig. 4 | Out-of-sample model EVI prediction accuracy: Sub-Saharan Africa.** Figure compares out-of-sample EVI predictions for piecewise linear regression models estimated using weather observations from CRU, ERA5-Land, and GMFD in Sub-Saharan Africa. Gray, blue, and yellow boxes show the percent reduction in root-mean-squared error (RMS) relative to a baseline model without weather variables. The gray boxes (first group) show results for a piecewise linear model estimated on monthly CRU data, where degree days are constructed using Thom's formula. The blue and yellow bars (second and third groups) show RMS reductions for models estimated on degree days from daily data provided by ERA5-land and GMFD, respectively. Within each group, the bars represent relationships between EVI and temperature estimated at the grid cell level for a data set including (i) grid cells that contain weather stations and (ii) grid cells without weather stations, indicated by box shading. For each data set, piecewise linear response functions are estimated 1,000 times, each time randomly sampling 75% of the years from the panel. RMS is calculated based on each piecewise linear model's prediction of the remaining 15% of years. Boxes, horizontal lines, and points represent the inter-quartile range, median, and mean of RMS reductions from the 1,000 draws. All models include grid-cell fixed effects, quadratic precipitation controls, and country-level quadratic time trends.

sets vary in terms of out-of-sample performance in Sub-Saharan Africa, differences due to the proximity of weather stations are small relative to overall performance, providing suggestive evidence that these data sets are able to estimate robust weather relationships in relatively data poor regions of the world.

## Discussion

Several studies have highlighted the importance of extreme temperatures on agricultural yields using fine-scaled data. Recently, two global data sets have become available that are temporally fine-scaled (providing daily minimum and maximum temperature or even hourly temperature values), but spatially more aggregated. The effect of temperature extremes might be masked if the weather data are aggregated, both temporally (a monthly average hides peak exposure within a month) as well as spatially (aggregating over larger areas can hide that some part of it experienced extremes).

While these global data sets have been used, there has not been a systematic assessment of whether they uncover the non-linearities and crucial effects of temperature extremes. Our results suggest that novel climate data sets with daily observations are useful additions for estimating global crop yield response functions, particularly if the objective is to simulate impacts of future climate scenarios. All three data sets correctly capture the harmful effects of temperature extremes

when linked to US corn and soybean yields. When we aggregate the data to seasonal averages, the common approach undertaken in previous global assessments that relied on monthly weather measures that cannot adequately capture daily temperature extremes, the out-of-sample predictive performance is greatly reduced. In fact, any of the three data sets using any of the three specifications using daily data perform better than any specification using monthly averages. The provision of daily global temperature data is hence an important step forward to accurately simulate the effects of climate change on agricultural yields and prices.

Differences in predictive skill may be driven by measurement error induced by a combination of the spatial and temporal resolution of the data sets as well as the interpolation methods used to fill gaps in the weather record. Supplementary Fig. 1 further examines the role of spatial and temporal aggregation. A "resampled PRISM model" estimated after spatially aggregating PRISM to the ERA5-Land resolution before constructing our county panel reduces the gap in RMS between the PRISM and ERA5-Land data sets by about 55%. In other words, half of the lead PRISM has over ERA5-Land in predicting yields is caused by the fact that it is spatially more disaggregated. The remaining gap in RMS is likely due to the input weather observations underlying the gridded products and the interpolation models (or "reanalysis") used to fill in gaps in the weather record across time and space. Reanalysis uses outputs of numerical weather prediction models to create a globally consistent record, which, in the process, may mask the most extreme weather events that are critical for accurately predicting yields. PRISM and GMFD reanalyze observations from on-the-ground weather stations. ERA5-Land, on the other hand, does not rely on observations from weather stations. Instead, it uses remotely sensed measurements of near-surface temperature and reanalysis to construct a global weather record[21].

Supplementary Fig. 1 also assesses the impact of differences in the temporal scale of the climate data. Temporally aggregating ERA5-Land from hourly temperature readings to daily minimum and maximum temperature before calculating degree days using a sine interpolation between minimum and maximum temperature negligibly reduces RMS, suggesting that the sub-daily hourly records do not provide important additional information. Taken together, a global data set on minimum and maximum temperature correctly identifies the importance of temperature extremes and provides yield predictions that closely mirror more fine-scaled weather data.

One qualification of our findings is that some global weather data sets, such as GMFD, assimilate the weather station network of a country, which is dense in the United States. Other countries have fewer stations and the global weather data sets accordingly might be worse at capturing temperature extremes. Figure 4 explores this issue by comparing yield predictions in locations with and without weather stations in Sub-Saharan Africa. We find that ERA5-Land, which relies on satellite measurements rather than weather station networks, is unsurprisingly unaffected by the proximity of weather stations. GMFD, which incorporates station data, shows a slight, but statistically significant, increase in predictive power when we subset our data to only locations near weather stations. However, this increase is small relative to the overall reduction in RMS associated with weather variables provided by GMFD, providing suggestive evidence that GMFD is also well-suited for estimating yield impacts in Sub-Saharan despite low weather station density relative to the United States.

While this analysis focuses specifically on yield relationships and climate projections of yield losses, our results provide insight for a broad set of topics within agricultural economics. For example, recent assessments of the global agricultural trade implications of climate change rely on spatial correlations of agricultural productivity under climate change scenarios e.g., ref. 22. We show that recently available global climate data products are able to estimate future agricultural productivity and support general equilibrium analyses that in turn

inform optimal climate change policy. In addition, our analysis suggests that globally consistent weather data products may be useful for research and application of index-based or farm-level insurance programs, which require accurate weather estimates and predictions of yield losses due to extreme events[23]. Finally, whereas yield data from administrative records have thus far been the limiting factor for estimating high resolution crop-weather response functions, increasing availability of farm-level yield data e.g., ref. 24 or yield data inferred from satellite based measurements[23,25] necessitate high-resolution climate data for analysis. While our study suggests that county-level relationships are stable across climate data sets despite differences in the scale of the raw data, future research might consider the implications of different spatial resolutions of the observational unit on this comparison.

## Concluding remarks

We find that two global data sets, ERA5-Land and GMFD, that provide daily temperatures uncover non-linear relationships between extreme heat and US crop yields that closely align with the results that are obtained using the high resolution PRISM data set. In particular, we find similar effects of extreme temperatures on both corn and soybean yields across three specifications and the three data sets. On the other hand, the benefits of yield-maximizing temperatures vary somewhat between models: the estimated relationship is lower under ERA5-Land than the other two climate data sets. Predictive power, in terms of RMS, is lower for the global data sets than for PRISM, with GMFD and ERA5-Land performing about the same. Despite these differences, projections of climate impacts from spatially uniform warming are similar across models that include daily temperature extremes. Finally, we validate the benefits of daily weather observations in the global data sets. However, using average temperatures over the growing season rather than accounting for daily extremes universally lowers predictive power and simulated climate impacts. The appropriate functional form is more important than the choice of weather data set.

## Methods
### Weather data

The four climate data sets used for this analysis are (1) a modified version of the Parameter-elevation Regressions on Independent Slopes Model (PRISM) data set from the Northwest Alliance for Computational Science and Engineering based at Oregon State University[1] used in our study of US yields; (2) the fifth generation of the European Reanalysis (ERA5-Land) data set from the European Center for Medium-Range Weather Forecasts; (3) the Global Meteorological Forcing Dataset (GMFD) from the Terrestrial Hydrology Research Group at Princeton University; and (4) the CRU TS v 4.07 data by the Climatic Research Unit of the University of East Anglia in our study of Sub-Saharan yields.

### Weather data—US analysis

For each grid cell, we approximate the distribution of temperatures within each day using a sinusoidal curve fit between minimum and maximum temperature measurements[26], except for ERA5-Land, which provides hourly measures (see below). We calculate two measures of daily temperature exposure. The first is the amount of time a pixel is exposed to each 1 °C interval each day. The second is an estimate of degree days, which measures for how long and by how much temperatures exceed a threshold. For example, for a threshold of 30 °C, a hypothetical day of constant 32 °C temperature contributes 2° days, as would 2 days at 31 °C, while a day of constant 28 °C temperature contributes 0 degree days. Note that in order to preserve non-linearities in the temperature record, it is important to construct daily temperature measurements at each weather grid cell before aggregating measurements to the county level. All daily grid cell-level temperature and precipitation measurements are combined with a high

resolution cropland raster that allows us to weight cells based upon the share of cropped area. We obtain high resolution cropland information from the USDA National Agricultural Statistics Service Cropland Data Layer (CDL). CDL provides crop-specific land cover masks for the continental US. Data are available annually from 2008 to 2021. We take an average of all available cropland rasters at the native 30 meter resolution before aggregating to the appropriate climate grid cell size. Weather data are linked to counties in the contiguous US in two ways: for the fine-scaled PRISM data we link a cell to county if its centroid falls within the county. For the spatially coarser GMFD and ERA5-Land data sets, we derive the fraction of a grid cell that overlaps with the county boundary as described next in more detail:

PRISM provides daily minimum temperature, maximum temperature, and total precipitation data on a 4 × 4 km grid across the contiguous United States. To stay consistent with the other weather sets, we focus on the period 1950–2019. The raw PRISM data are modified by ref. [1] to maintain a constant set of stations over time, ruling out that changes in temperatures are due to a change in station coverage. To obtain a balanced panel of weather observations, missing weather measurements are filled in with the distance-weighted average of the cumulative density function of surrounding stations. For example, if the 10 closest weather stations are on average at their 70th percentile, the station's missing value is set to the 70th percentile of its own measurements. We link a PRISM cell to the county in which its centroid is located.

ERA5-Land provides average hourly temperature and total precipitation measurements on a global 0.1° ($\approx$ 11.1 km) grid for the period 1950–2019. The process of aggregating these data to the county level is similar to the process applied to PRISM with two exceptions. First, hourly temperature data allow us to calculate daily temperature exposure directly instead of interpolating the within-day distribution using a sinusoidal curve. Hourly data should produce temperature variables with less measurement error than those derived from PRISM. In Supplementary Fig. 1, we assess the degree to which this impacts model performance by applying the sinusoidal fit to the minimum and maximum daily temperatures from ERA5-Land and comparing out-of-sample predictions. We find that models estimated with interpolated ERA5-Land data perform negligibly worse than those based on the raw hourly data. Second, we aggregate weather observations to counties weighted by cropped area as well as the share of each grid cell that intersects the county. Thus, a cell that only partially overlaps a county's boundary is weighted less than one that lies entirely within the county.

GMFD provides daily minimum temperature, maximum temperature, and total precipitation data on a global 0.25 × 0.25 degree ($\approx$ 28 km) grid for the period 1950–2010. GMFD observations are aggregated to the county level using a combination of the methods described above. We use a sinusoidal interpolation of within-day observations to measure temperature exposure, and we include area weights in the spatial aggregation to account for the low resolution of the data. Visual comparisons of county-level climate data are provided in Supplementary Figs. 4 and 5.

## Weather data—Sub-Saharan Africa analysis

The spatial scale for the Sub-Saharan analysis is the ERA5-Land grid raster. For the coarser GMFD and CRU grids, we assign their values to all ERA5-Land grids they contain. We subset each climate data set to grid cells containing corn crops based upon a satellite scan from the FAO Global Agro-Ecological Zones (GAEZ). We construct degree day measures for Sub-Saharan African grid cells, weighted by corn cropped area, for the three data sets. Degree day measures for ERA5-Land and GMFD are calculated as described in the US section above. For CRU, which only provides monthly observations, we estimate degree days using Thom's formula[27]. We aggregate the weather data over the subregion-specific growing season definitions from ref. [28]. We then incorporate data from the National Climate Data Center (NCDC) to identify the grid cells that contain weather stations.

## Yield data

Yield data are collected from the U.S. Department of Agriculture's National Agricultural Statistical Service. We use corn and soybean yields from the years 1950 to 2019 for the main analysis; however, 9 years are dropped for regressions with GMFD, which only provides a weather record through 2010. We focus on drylands yields, i.e., counties east of the 100° meridian and excluding Florida, to avoid bias associated with subsidized irrigation. When merged with the weather data, our panel consists of 128,169 observations for corn yields and 102,674 observations for soy yields, with approximately 14,000 observations dropped for regressions on GMFD weather variables.

Since we do not have sub-national yield data for Sub-Saharan Africa, we use the monthly Enhanced Vegetation Index (EVI) data from Landsat to proxy for yields for that analysis. Similar to the weather data described above, we mask out grid cells without crops based upon the GAEZ satellite data. We calculate the sum of EVI over the corn growing season in each grid cell based upon subregion-specific growing season definitions from ref. [28].

## Regression models—US

Following ref. [1], we model the relationship between weather and yields assuming that the effect of increasing temperatures on yields is additively separable over the growing season. For example, our models assume that an additional degree day experienced just after planting has the same effect on yields as an additional growing degree day experienced right before harvest. Our model of plant growth $g(h)$ is from ref. [1] and is a nonlinear function of heat $h$. Thus, log yield $y_{it}$ in county $i$ and year $t$ is

$$y_{it} = \int_{\underline{h}}^{\overline{h}} g(h)\phi_{it}(h)dh + z_{it}\delta + c_i + \epsilon_{it}. \tag{1}$$

$\phi_{it}(h)$ is the time distribution of heat over the growing season for county $i$ and year $t$; $\underline{h}$ and $\overline{h}$ are the lower and upper bounds of temperatures observed over the growing season; $z_{it}$ is a vector of additional time-varying controls, including a quadratic function of total growing season precipitation and state-specific quadratic time trends, which account for technological progress common to all counties within a state; $c_i$ are county fixed effects, which flexibly control for time-invariant characteristics of counties that confound the weather-yield relationship, such as soil quality. Our preferred specification clusters standard errors at the state level. Supplementary Datasets 1 and 2 provide results under alternative specifications, including year fixed effects and year-clustered and Conley standard error assumptions. Note that the inclusion of fixed effects in our model implies that relationships should be interpreted in relative terms. As such, we normalize response functions in our plots to an arbitrary value. Specifically, the temperature responses provided in Fig. 1 are normalized to 10 °C and the precipitation responses in Supplementary Fig. 2 are normalized to 0 cm of rainfall.

We focus on the months March through September to capture the main growing season for both corn and soybeans. Again following the ref. [1] and using the temperature constructions described above for each 1 °C temperature interval, we approximate (1) with

$$y_{it} = \sum_{h=-4}^{41} g(h+0.5)\big[\Phi_{it}(h+1) - \Phi_{it}(h)\big] + z_{it}\delta + c_i + \epsilon_{it} \tag{2}$$

where $\Phi_{it}(h)$ is the cumulative distribution function of heat exposure in county $i$ and year $t$. We model $g(h)$ with three functional form assumptions: a piecewise linear function estimated via degree days at a crop-specific threshold temperature, an eighth order Chebychev polynomial, and a non-parametric specification in which a separate yield effect is estimated for each 3 °C temperature bin up to 36 °C.

We compare out-of-sample performance and climate projections from the daily responses to functions of temperature that rely on

monthly averages over the growing season. For the monthly responses, we estimate:

$$y_{it} = T_{it} + T_{it}^2 + z_{it}\delta + c_i + \epsilon_{it} \qquad (3)$$

where $T_{it}$ is the monthly average temperature (i.e., we collapse the hourly or daily data to monthly averages) averaged again over the growing season (March-September) in county $i$ and year $t$. Quadratic precipitation controls, county fixed effects, and state-specific quadratic time trends are consistent with the daily model.

### Regression models—Sub-Saharan Africa

Since the EVI data is only available since 2000 and since GMFD ends in 2010, we are left with a panel data for 11 years, 2000–2010. Furthermore, since CRU only provides monthly values and not daily values, we use Thom's formula[27] to approximate degree days and focus on the piecewise linear model only. Recall from above weather data section that the spatial unit is the ERA5-Land grid and we aggregate the weather and EVI data over the subregion-specific growing season definition for corn from ref. 28.

Our data consists of about 103,000 grid cells with only about 1200 grid cells containing weather stations. Weather stations are not uniformly distributed across countries in Sub-Saharan Africa. For example, about 70% of the stations in the NCDC are in Namibia and South Africa alone. If there is heterogeneity in the crop yield-weather relationship across countries in Sub-Saharan Africa, then an analysis that only considers grid cells with weather stations will over sample these countries relative to an analysis that only considers grid cells without stations. To avoid this bias in our out-of-sample comparison, we repeatedly take a stratified random sample of grid cells without weather stations based upon the number of weather stations present in each country as defined by the NCDC data. Thus, our sample of 1200 grid cells with stations is compared to a random sample of 1200 grid cells without stations, where the composition of countries within each data set is similar.

In particular, we repeat the following procedure separately for grid cells with and without weather stations. For each data set, we estimate grid-cell level piecewise linear relationships between the log of total EVI over the growing season and temperature on 75% of the years from the full panel (2000–2010) 1000 times. Since there are many more grid cells without weather stations, each of the 1000 iterations randomly samples grid cells from each country according to the number of stations present in the NCDC data. For each iteration, RMS is calculated for the grid cells included in the regression, but for years excluded from the regression relative to a baseline model with grid cell fixed effects and country-level quadratic time trends. Models for each weather data set also include quadratic precipitation controls. Note that the bounds for the piecewise linear function are taken from the main analysis and not re-estimated endogenously.

### Reporting summary

Further information on research design is available in the Nature Portfolio Reporting Summary linked to this article.

## Data availability

The analysis relies on public use weather data from PRISM, ERA5-Land (https://cds.climate.copernicus.eu/) and GMFD (https://rda.ucar.edu/datasets/ds314.0/), as well as data on agricultural production from U.S. Department of Agriculture's National Agricultural Statistical Service (https://www.nass.usda.gov/Data_and_Statistics/). We also rely on cropland data layers from USDA and FAO GAEZ as well as weather station locations from National Climate Data Center (https://www.ncei.noaa.gov/cdo-web/). Spatial aggregation of climate data and maps produced in the Supplementary Information rely on shapefiles provided by the US Census Bureau (https://www.census.gov/geographies/mapping-files/time-series/geo/tiger-line-file.html). The cleaned and aggregated data used to produce the analysis in this study and the Supplementary Information have been deposited in the Harvard Dataverse repository under a Creative Commons 4.0 license (https://doi.org/10.7910/DVN/XRMDBW).

## Code availability

The code developed to produce the analysis in this paper and the Supplementary Information is available in a Github repository[29] under a GNU General Public License v3.0 along with instructions for replicating the results.

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

## Acknowledgements

This work was supported by USDA National Institute of Food and Agriculture, Grant No. 2022-67023-36400 (D.H. and W.S.) as well as by the US Department of Energy, Office of Science, Biological and Environmental Research Program, Earth and Environmental Systems Modeling, MultiSector Dynamics, Contract No. DE-SC0016162 (D.H. and W.S.).

## Author contributions

D.H. and W.S. designed the study, collected the data, performed the analysis and wrote the paper.

## Competing interests

The authors declare no competing interests.
