## [Peer Review File · Nature Communications]

Reviewers' Comments:

Reviewer #1:

Remarks to the Author:

This is a nice, tight paper with a clear focus on assessing weather data quality for an important subset of global-gridded products. The work is much needed, and represents an important step forward in gaining credibility for broader research linking climate to agricultural and economic outcomes at the global scale. My comments focus on improving the messaging of the paper, as I view the methods as being sound and the results are correctly drawn from the analysis. None of these comments are fatal, but I do think they will help readers engage with the paper.

1. Overall I don't think the paper is doing enough to engage with the existing literature. Here are some thoughts, which could be used to enhance communication in the introduction and/or conclusions.

A. Ties to the index insurance literature can be improved. Lobel and others have some recent contributions in the AJAE and other journals. It's not clear that these products could improve basis risk problems, but they would permit farm-level insurance product rollouts at large spatial scales. And then there are the corporate agents operating in the re-insurance markets. Ben Collier has some papers that could be relevant. Same with Robert Finger.

B. Ties to the trade literature. If we are to understand how trade patterns might alter in response to climate change, these data products are essential. Nelson Villoria, Sol Hsiang, and others have some papers in this space.

C. There is a growing literature linking weather to crop yields at the farm/field trial level. These are important studies for better understanding behavioral adaptation, both on-farm management decisions and field-trial assessments of genetic potential. This literature could be referenced, along with a short discussion on how this analysis might inform such studies. Perhaps this is best cast as an area for future research to consider. It would be interesting to see if these data products could be useful for estimating response curves at the farm/field level. AJAE has some recent papers using farm data. Ariel Ortiz Bobea's recent Handbook might have some recent references.

2. There are a few statements that require some empirical evidence to support. Suggest including tables/figures in the supp appendix to support.

A. Similar U-shaped precip effects (lines 146-148)

B. Similar effects with year FE (lines 150-52)

C. Some counties experience yield benefits (lines 192-93). Suggest including spatial maps of the warming effects at the grid-cell or county level. Focus on +2C and include one map for each crop-dataset combination.

Other Comments

Yield instead of Yields on line 67

Italics on lines 95-96?

Note that the CIs account for spatial correlation when first mentioned, line 109 I think

Please be explicit on how the response curves are normalized (lines 123-24). Can include a couple sentences in the Methods.

Footnote 2 is important. Readers will be curious why GCMs aren't being used to assess the similarity of results. I think the uniform warming impacts are very insightful, so I am not suggesting you change that; just bring the reasoning for the focus into the main verbiage and not bury it in a

footnote.

Lines 308-313. Im guessing the focus is on dryland yields? Are there lat/lon restrictions on the sample?

Methods Section. Some clear similarities to the Schlenker Roberts 2009 PNAS paper. Thats all good, but be careful about replicating exact verbiage without referencing. Or include some blanket statement making clear where the methods are same/different.

Line 326. Clustering at state level is likely to be too restrictive. Suggest clustering at year level, or use the Conley spatial-decay SEs.

After line 334. Suggest including an equation for the avg temp model so readers are clear how that is being specified. Im guessing quadratic. If linear should probably be made quadratic.

Reviewer #2:

Remarks to the Author:

This is an interesting and important paper which compares the explanatory and predictive power of three different high resolution weather datasets on US corn and soybean yields. The authors compare the current "best practice" high resolution PRISM dataset from Oregon State University to two newer reanalysis datasets (ERA5-Land and GMFD).

There are a few really important findings. The authors show that for the US, both "new" weather datasets allow one to recover the non-linear relationships between temperature and yields that have been confirmed using station and PRISM data before. This means that these reanalysis data do have enough signal in them to estimate these relationships. One critique of these models is that they introduce measurement error, which if classical, would lead to attenuated relationships.

This matters as PRISM exists for the US only and similar datasets are not available globally. This would hence lead one to hope that these global datasets would allow us to better understand the weather yield relationships in areas with worse station coverage as well as for different crops and potentially outcomes (e.g. mortality, energy usage).

The work is clearly important, and significant to the literature modelling climate damages. The idea behind this paper is original. Instead of comparing weather datasets, they compare predictive ability, which is novel. The conclusions are well supported and clearly stated. The methodology is sound and represents best practice at the frontier of statistical climate impacts modelling. I did not review any code, but think I could replicate these results from the description in the paper.

I have the following comments:

- There is a bit of a chicken and an egg problem here. Areas with better coverage of weather stations, will also arguably be the areas where the reanalysis models do better, as they have more data to work with for parametrization. Hence, using the United States with the world's best weather station data coverage makes these reanalysis data pretty accurate. But the big benefits here, as the authors point out, would come from places with sparse data coverage. Like sub Saharan Africa. The authors state this in the paper, but I still wonder if one could do a little bit more. I note that one of the authors has a paper looking at crop yield impacts in Africa. Would it be possible to see whether they can replicate those findings using these data? One could just use the gridpoints, which would prevent a major data exercise.
- The paper early on makes a statement about weather impacts needed for climate impacts estimation. This is a contentious point in the literature. While I understand where the authors are coming from, it is important to acknowledge that a relationship calibrated based on short run weather fluctuations should probably not be used for long run climate damage estimation as the function changes due to adaptation. To be clear, I am not asking the authors to do more here, but the framing is important. I am referring to lines 26/27.
- A minor point. Data are plural, not matter what "The Economist" says. Please change this to

plural throughout (e.g. line 34).

- There is a point made about measurement error that I think needs some clarification. I am referring to line 46. I understand how classical measurement error leads to attenuation. But I am not sure that is true for all types of measurement error. Maybe I am being picky here, but some additional clarification or a citation or two would really help me think this through. The example given does some of that.
- Line 67. It should be "yield data" not "yields data"
- Line 72. It would be good to state here that the crop coverage mask does not go very far back in time.
- I am having trouble understanding the critical discussion of the CMIP model output versus the uniform warming scenarios. Warming is clearly not uniform. I understand why you would do this as a proof of concept. I would probably just drop the CMIP discussion or rewrite it, so it is easier to understand.
- The graphs are really beautiful. I would advocate for color schemes for folks that cannot differentiate some of the colors. I will leave it up to the editor to see whether this is necessary, but it helps folks like me.
- I was somewhat surprised that no attempt is made to compare the weather data themselves across the datasets. Since the authors (I think) use a modified version of PRISM, a citation to other work of theirs that shows a comparison would work, or some scatters or regressions (or overlaid histograms).
- Line 161. Do you select random combinations of single years for all counties? Or county/years? I think it is just years. The justification for this is stated, but I do not fully understand the pointer at spatial correlation (since the standard errors are clustered, this should not matter I think).
- Cite the Welch Test.

ERA5-Land and GMFD Uncover The Effect of Daily Temperature Extremes on Agricultural Yields
Dylan Hogan and Wolfram Schlenker
Responses to Reviewer 1

Reviewer #1 (Remarks to the Author):

This is a nice, tight paper with a clear focus on assessing weather data quality for an important subset of global-gridded products. The work is much needed, and represents an important step forward in gaining credibility for broader research linking climate to agricultural and economic outcomes at the global scale. My comments focus on improving the messaging of the paper, as I view the methods as being sound and the results are correctly drawn from the analysis. None of these comments are fatal, but I do think they will help readers engage with the paper.

1. Overall I don't think the paper is doing enough to engage with the existing literature. Here are some thoughts, which could be used to enhance communication in the introduction and/or conclusions.

A. Ties to the index insurance literature can be improved. Lobel and others have some recent contributions in the AJAE and other journals. It's not clear that these products could improve basis risk problems, but they would permit farm-level insurance product rollouts at large spatial scales. And then there are the corporate agents operating in the re-insurance markets. Ben Collier has some papers that could be relevant. Same with Robert Finger.

B. Ties to the trade literature. If we are to understand how trade patterns might alter in response to climate change, these data products are essential. Nelson Villoria, Sol Hsiang, and others have some papers in this space.

C. There is a growing literature linking weather to crop yields at the farm/field trial level. These are important studies for better understanding behavioral adaptation, both on-farm management decisions and field-trial assessments of genetic potential. This literature could be referenced, along with a short discussion on how this analysis might inform such studies. Perhaps this is best cast as an area for future research to consider. It would be interesting to see if these data products could be useful for estimating response curves at the farm/field level. AJAE has some recent papers using farm data. Ariel Ortiz Bobea's recent Handbook might have some recent references.

Thank you very much for this comment. We added a paragraph discussing these ties to the literature in the discussion section (lines 344-359):

"While this analysis focuses specifically on yield relationships and climate projections of yield losses, our results provide insight for a broad set of topics within agricultural economics. For example, recent assessments of the global agricultural trade implications of climate change rely on spatial correlations of agricultural productivity under climate change scenarios (e.g., 7). We show that recently available global climate data products are able to estimate future agricultural productivity and support general equilibrium analyses that in turn inform optimal climate change policy. In addition, our analysis suggests that globally consistent weather data products may be useful for research and application of index-based or farm-level insurance programs, which require accurate weather estimates and predictions of yield losses due to extreme events (20). Finally, whereas yield data from administrative records have thus far been the limiting factor for estimating high resolution crop-weather response functions, increasing availability of farm-level yield data (e.g., 11) or yield data inferred from satellite based measurements (20; 26) necessitate high-resolution climate data for analysis. While our study suggests that county-level relationships are stable across climate data sets despite differences in the scale of the raw data, future research might consider the implications of different spatial resolutions of the observational unit on this comparison."

2. There are a few statements that require some empirical evidence to support. Suggest including tables/figures in the supp appendix to support.

A. Similar U-shaped precip effects (lines 146-148)

We added a new Section 2 to the supplementary materials called "Precipitation response functions", which includes plots of the precipitation effects with 95% confidence intervals (Supplementary Figure 2). In-text reference to these

figures is on line 176. Also note that the quadratic precipitation coefficients are provided in Supplementary Tables 1 and 2.

B. Similar effects with year FE (lines 150-52)

Supplementary Tables 1 and 2 compare model results under different fixed effect and standard error assumptions. Columns 4-6 of ST1 (Corn) and ST2 (Soybeans) provide coefficients for models with year fixed effects, under assumptions of state clustering, year clustering, and Conley standard errors, respectively, for each climate data set. In-text reference to the tables is on line 145.

C. Some counties experience yield benefits (lines 192-93). Suggest including spatial maps of the warming effects at the grid-cell or county level. Focus on +2C and include one map for each crop-dataset combination.

Supplementary Figure 3 shows maps of yield losses under 2°C warming for each crop-data set combination. High latitude counties experience slight yield benefits from 2°C warming for each crop/data set. We reference this figure on line 233.

Other Comments

Yield instead of Yields on line 67
We implemented this change on new line 77.

Italics on lines 95-96?

We implemented this change on new lines 120-121.

Note that the CIs account for spatial correlation when first mentioned, line 109 I think

We implemented this change on new lines 135-136.

Please be explicit on how the response curves are normalized (lines 123-24). Can include a couple sentences in the Methods.

Based on this suggestion, we modified Figure 1 to normalize all response functions at 10°C. Our original draft normalized curves at the PRISM exposure-weighted average temperature. We added a couple sentences referencing this in the main text (line 150) as well as the methods (line 462)

Footnote 2 is important. Readers will be curious why GCMs aren't being used to assess the similarity of results. I think the uniform warming impacts are very insightful, so I am not suggesting you change that; just bring the reasoning for the focus into the main verbiage and not bury in a footnote.

We received opposing comments about this particular language, where another reviewer suggested that we change or remove the footnote. We rewrote our discussion so that the intuition behind our decision to avoid GCM simulations is (hopefully) clearer and merged it into the main text of the paper, as you suggested. See lines 216-223:

"Many studies have used global climate models (GCMs) to project agricultural losses of spatially heterogeneous climate change scenarios, such as those from CMIP5 or CMIP6. However, errors in GCM output can be large relative to historical data sets (15) and must be bias corrected to a particular historical data set before climate impacts are calculated. There is also disagreement between GCM models on the magnitude of warming. We therefore opt for simple uniform warming scenarios that allow us to compare across data sets and to highlight the range of warming at which the models start to diverge."

Lines 308-313. I'm guessing the focus is on dryland yields? Are there lat/lon restrictions on the sample?

That is correct. We made this clearer in the text, with references in lines 435 in the main text and 78 in the Methods section. The longitude restrictions are all counties east of the 100° meridian, consistent with Schlenker and Roberts (2009).

Methods Section. Some clear similarities to the Schlenker Roberts 2009 PNAS paper. That's all good, but be careful about replicating exact verbiage without referencing. Or include some blanket statement making clear where the methods are same/different.

Thank you for the comment. We rewrote parts of this section and included more explicit references where necessary.

Line 326. Clustering at state level is likely to be too restrictive. Suggest clustering at year level, or use the Conley spatial-decay SEs.

As noted above, in Supplementary Tables 1 and 2, we provide coefficients for the piecewise linear model under a range of fixed effect and clustering assumptions. As you suggest, we include columns (2-3 for state trends; 5-6 for year fixed effects) which show year and Conley clustered standard errors.

After line 334. Suggest including an equation for the avg temp model so readers are clear how that is being specified. I'm guessing quadratic. If linear should probably be made quadratic.

The average temperature model is indeed quadratic. We made this clearer throughout the paper and included the equation for this model in lines 471-477.

ERA5-Land and GMFD Uncover The Effect of Daily Temperature Extremes on Agricultural Yields
Dylan Hogan and Wolfram Schlenker
Responses to Reviewer 2

Reviewer #2 (Remarks to the Author):

This is an interesting and important paper which compares the explanatory and predictive power of three different high resolution weather datasets on US corn and soybean yields. The authors compare the current “best practice” high resolution PRISM dataset from Oregon State University to two newer reanalysis datasets (ERA5-Land and GMFD).

There are a few really important findings. The authors show that for the US, both “new” weather datasets allow one to recover the non-linear relationships between temperature and yields that have been confirmed using station and PRISM data before. This means that these reanalysis data do have enough signal in them to estimate these relationships. One critique of these models is that they introduce measurement error, which if classical, would lead to attenuated relationships.

This matters as PRISM exists for the US only and similar datasets are not available globally. This would hence lead one to hope that these global datasets would allow us to better understand the weather yield relationships in areas with worse station coverage as well as for different crops and potentially outcomes (e.g. mortality, energy usage).

The work is clearly important, and significant to the literature modelling climate damages. The idea behind this paper is original. Instead of comparing weather datasets, they compare predictive ability, which is novel. The conclusions are well supported and clearly stated. The methodology is sound and represents best practice at the frontier of statistical climate impacts modelling. I did not review any code, but think I could replicate these results from the description in the paper.

I have the following comments:

- There is a bit of a chicken and an egg problem here. Areas with better coverage of weather stations, will also arguably be the areas where the reanalysis models do better, as they have more data to work with for parametrization. Hence, using the United States with the world’s best weather station data coverage makes these reanalysis data pretty accurate. But the big benefits here, as the authors point out, would come from places with sparse data coverage. Like sub Saharan Africa. The authors state this in the paper, but I still wonder if one could do a little bit more. I note that one of the authors has a paper looking at crop yield impacts in Africa. Would it be possible to see whether they can replicate those findings using these data? One could just use the gridpoints, which would prevent a major data exercise.

Thank you very much for this comment. We spent quite a lot of time considering this issue, and we hope the additional results we provide in the main text of the paper are sufficient. We added an analysis in which we evaluate the out of sample performance of the global data sets (ERA5-Land and GMFD) by comparing grid-cell level response functions in Sub-Saharan Africa. We encountered several challenges in the process. Full details are provided in the Results (line 243) and Methods (lines 420; 478) sections, but the following is a short summary.

First, weather station data we obtained from NCDC contained too many gaps in the weather record to be considered “truth” against which we evaluate the global data sets. To overcome this, we identify the location of weather stations that would be assimilated into the global data sets and compare predictive performance between response functions estimated on grid cells containing weather stations to those without weather stations. Thus, we evaluate whether the proximity of weather stations influences the out of sample performance of the responses, since grid cells with weather stations should, in theory, have more accurate weather measurements than grid cells far away from the closest ground measurement. We hope this comparison addresses the spirit of your question.

Second, we were unable to find reliable yield data to estimate our responses at the grid-cell level, which was necessary for our comparison. We overcame the second challenge by using Enhanced Vegetation Index (EVI) as a

proxy for yields, a strategy that is becoming increasingly common in economics and other disciplines that study agriculture. Please see the Supplementary Information for more information about how we use the EVI data and estimate responses.

In general, we find large reductions in RMS from including weather variables in our regressions (see Figure 4). ERA5-Land has particularly high out of sample performance, likely due to the fact that ground level temperature measurements are not assimilated into that data product (rather, only satellite observations of near-surface temperature). GMFD also performs statistically significantly better than CRU, which is a global dataset providing interpolated weather measurements and is among the inputs to GMFD. Reductions in RMS from focusing only on grid cells without weather stations are small relative to the overall predictive performance of our models. Differences for ERA5-Land and CRU are not statistically significant, while GMFD models estimated on weather station cells achieve significantly better performance.

- The paper early on makes a statement about weather impacts needed for climate impacts estimation. This is a contentious point in the literature. While I understand where the authors are coming from, it is important to acknowledge that a relationship calibrated based on short run weather fluctuations should probably not be used for long run climate damage estimation as the function changes due to adaptation. To be clear, I am not asking the authors to do more here, but the framing is important. I am referring to lines 26/27.

Thanks for this suggestion. We made changes to the language of the opening paragraph (new lines 31-41) and included a footnote (new FN1) to make this distinction clear.

- A minor point. Data are plural, not matter what "The Economist" says. Please change this to plural throughout (e.g. line 34).

We implemented this change throughout the paper.

- There is a point made about measurement error that I think needs some clarification. I am referring to line 46. I understand how classical measurement error leads to attenuation. But I am not sure that is true for all types of measurement error. Maybe I am being picky here, but some additional clarification or a citation or two would really help me think this through. The example given does some of that.

Thank you for this feedback. As you point out, classical measurement error would attenuate our results, but the potential types of measurement error induced by differences in climate data sets (including resolution, interpolation or reanalysis, etc.) may not be classical in nature. This discussion is trying to make a point specifically about the spatial resolution of the data, i.e. what happens when you average temperature measurements over space. We tried to make that clearer in the discussion (see new lines 51-61) and also added a citation to a review paper where this issue is discussed (Hsiang, 2016).

- Line 67. It should be "yield data" not "yields data"

We implemented this change on new line 77.

- Line 72. It would be good to state here that the crop coverage mask does not go very far back in time.

We implemented this change on new lines 83-84.

- I am having trouble understanding the critical discussion of the CMIP model output versus the uniform warming scenarios. Warming is clearly not uniform. I understand why you would do this as a proof of concept. I would probably just drop the CMIP discussion or rewrite it, so it is easier to understand.

We received opposing comments about this particular language, where another reviewer suggested that we bring this footnote into the main text of the paper. We rewrote the discussion so that the intuition is (hopefully) clearer. See lines 216-223::

"Many studies have used global climate models (GCMs) to project agricultural losses of spatially heterogeneous climate change scenarios, such as those from CMIP5 or CMIP6. However, errors in GCM output can be large relative to historical data sets (15) and must be bias corrected to a particular historical data set before climate impacts are calculated. There is also disagreement between GCM models on the magnitude of warming. We therefore opt for simple uniform warming scenarios that allow us to compare across data sets and to highlight the range of warming at which the models start to diverge."

- The graphs are really beautiful. I would advocate for color schemes for folks that cannot differentiate some of the colors. I will leave it up to the editor to see whether this is necessary, but it helps folks like me.

Thanks for this comment. We opted to change the color scheme based on a color blind-safe palette. We took the palette from this source: <https://personal.sron.nl/~pault/#sec:qualitative>. In particular, we use colors Red, Blue, Green from their Figure 1. Please let us know if the colors continue to be confusing; we are happy to change them again based on your or the editor's guidance.

- I was somewhat surprised that no attempt is made to compare the weather data themselves across the datasets. Since the authors (I think) use a modified version of PRISM, a citation to other work of theirs that shows a comparison would work, or some scatters or regressions (or overlaid histograms).

We added Section 5 to the Supplementary Information comparing the weather data across data sets. In particular, Supplementary Figure 4 shows pairwise comparison scatter plots of the (demeaned and detrended) temperature variables across the three data sets. This figure also shows trend lines and partial correlations. Supplementary Figure 5 provides maps of growing season temperature, both degree days and average temperature, at the county level on average over the sample for each climate data set. These figures are referenced on 419.

- Line 161. Do you select random combinations of single years for all counties? Or county/years? I think it is just years. The justification for this is stated, but I do not fully understand the pointer at spatial correlation (since the standard errors are clustered, this should not matter I think).

We randomly draw years, so we estimate relationships using all counties in 85% of the years in the sample and then predict yields in all counties in the other 15% of years. While we cluster standard errors to account for spatial correlation in the estimated relationships, the clustered standard errors are not relevant for the out of sample prediction comparison, as this comparison is based on (point estimate) predictions of yields in years that were not used to estimate the weather-yield relationships.

If we sampled observations (i.e., county/year) rather than years, then counties left out of the estimation sample likely have neighbors with similar yields and temperature exposure in that year that remain in the estimation sample, which would lead to biased estimates of out-of-sample fit (RMS). On the other hand, year-to-year fluctuations in temperature shocks are random and therefore do not enter into the estimation sample alone, giving more reliable estimates of out-of-sample predictive skill.

- Cite the Welch Test.

We implemented this change on new line 201.

Reviewers' Comments:

Reviewer #1:

Remarks to the Author:

The authors have addressed my concerns. Great paper.

Reviewer #2:

Remarks to the Author:

I really appreciate the hard work that went into this significant revision. I only have the following minor comments. This paper will be very important to the climate impacts community.

Congratulations.

Line 35: "from the impacts" should be "for the impacts"

Line 164: The line "Substituting a full day at 10°C with a full day at 40°C decreases corn yields between 3.2% and 3.8% (SE 1.0%-1.2%) for all three data sets", seems like a crazy large counterfactual. I am not suggesting you change it, but it stuck out while reading the manuscript.

Line 245: I apologize for making you jump through very tall hoops with this Africa exercise. I appreciate the significant work and thoughtfulness of the approach that went into it. I read through it a few times and think this adds a lot to the paper. Thank you for doing this.

Line 329: "Assess" should be "assesses"

Line 583: I think the Tack reference is duplicated (21 & 22)

Figure 2: It took me two or three looks to get that the shading tells me functional form, and the color the dataset. Might be fine, but was not obvious right away, as the scale/legend is in gray and the bars are in color.